# FilamentSensor 2.0: An open-source modular toolbox for 2D/3D cytoskeletal filament tracking

Lara Hauke[1,2,3]*, Andreas Primeßnig[1,2], Benjamin Eltzner[4,5], Jennifer Radwitz[1,6], Stefan F. Huckemann[5], Florian Rehfeldt[1,7]*

1 Third Institute of Physics—Biophysics, Georg-August-University Göttingen, Göttingen, Germany, 2 Institute of Pharmacology and Toxicology, University Medical Center, Göttingen, Germany, 3 CIDAS (Campus Institute Data Science), University of Göttingen, Göttingen, Germany, 4 Research Group Computational Biomolecular Dynamics, Max Planck Institute for Multidisciplinary Sciences, Göttingen, Germany, 5 Felix-Bernstein-Institute for Mathematical Statistics in the Biosciences, Georg-August-University Göttingen, Göttingen, Germany, 6 Department of Molecular Neurogenetics, ZMNH, University Medical Center Hamburg-Eppendorf, Hamburg, Germany, 7 Experimental Physics I, University of Bayreuth, Bayreuth, Germany

* lara.hauke@med.uni-goettingen.de (LH); florian.rehfeldt@uni-bayreuth.de (FR)

**Data Availability Statement:** All supplemental files, the software and microscopy images are publicly available: https://doi.org/10.5281/zenodo.7314056.

## Abstract

Cytoskeletal pattern formation and structural dynamics are key to a variety of biological functions and a detailed and quantitative analysis yields insight into finely tuned and well-balanced homeostasis and potential pathological alterations. High content life cell imaging of fluorescently labeled cytoskeletal elements under physiological conditions is nowadays state-of-the-art and can record time lapse data for detailed experimental studies. However, systematic quantification of structures and in particular the dynamics (i.e. frame-to-frame tracking) are essential. Here, an unbiased, quantitative, and robust analysis workflow that can be highly automatized is needed. For this purpose we upgraded and expanded our fiber detection algorithm FilamentSensor (FS) to the FilamentSensor 2.0 (FS2.0) toolbox, allowing for automatic detection and segmentation of fibrous structures and the extraction of relevant data (center of mass, length, width, orientation, curvature) in real-time as well as tracking of these objects over time and cell event monitoring.

## Introduction

The cytoskeleton is a main contributor in many biological cellular processes that are governed by mechanics like mitosis and migration and is in general essential to maintain cellular homeostasis [1, 2]. Thus, a deeper understanding of the cytoskeleton structure and its dynamics (e.g. turnover, modulation, termination, and branching events) [3, 4] is key to understand the fundamental processes involved in cell- and matrix-mechanics.

While the detailed list of cytoskeletal key players and constituents involved is well known for the past decades [3, 5], emergent phenomena, in particular in complex environments (such as the mechanical coupling of cells with their surroundings, i.e. via focal adhesions [6], and downstream effects) are still not fully understood.

**Funding:** FR 491183248, Deutsche Forschungsgemeinschaft, https://www.dfg.de/en/index.jsp FR Open Access Publishing Fund of the University of Bayreuth, https://www.ub.uni-bayreuth.de/en/digitale_bibliothek/open_access/index.html FR SFB 755, B08, Deutsche Forschungsgemeinschaft, https://www.dfg.de/en/index.jsp FR, LH SFB 937, A13, Deutsche Forschungsgemeinschaft, https://www.dfg.de/en/index.jsp SH SFB 755, B08, Deutsche Forschungsgemeinschaft, https://www.dfg.de/en/index.jsp SH SFB 1456, A01, Deutsche Forschungsgemeinschaft, https://www.dfg.de/en/index.jsp BE SFB 1456, B02, Deutsche Forschungsgemeinschaft, https://www.dfg.de/en/index.jsp The funders had no role in study design, data collection and analysis, decision to publish, or preparation of the manuscript.

Previous attempts to quantify cytoskeletal structures were mostly limited to individual parameters or used a pixel-based method to extract global cellular descriptors [7]. Most importantly, they only focused on one particular constituent (e.g. actin or microtubules) of the cytoskeleton. However, the cellular cytoskeleton is more than the sum of its parts (actin fibers, microtubules, and the family of intermediate filament proteins) and only a holistic analysis will eventually lead to a full understanding of pattern formation and emerging behaviors in the cell. Additionally, due to broad distribution of cytoskeletal conformations and events, a large sample number and robust and automatized quantification of all accessible data is needed to discern between internal variability and genuine effect.

In the past, pattern formation of the cytoskeleton, e.g. mechano-guided differentiation in human mesenchymal stem cells (hMSCs) in response to varying matrix elasticity, has been explored in terms of an order parameter of stress fibers [7, 9–11]. Similar parameters have been used to describe the entirety of the microtubule network during cell cycle [12]. In contrast, small scale changes as in the early onset of cancer [13, 14] still need to be investigated. For these patterns, which can be easily overlooked in heterogeneous samples or variable data sets, a tool for fast and robust unsupervised analysis of unbiased data sets is needed to differentiate effects.

Previously, analysis of stress fibres was done manually or semi-automatically, limiting the experimental throughput and maximal data size. However, the amount of data recorded by novel high-content imaging systems providing better resolution at high frame rates renders manual annotation impossible. Large scale unsupervised analysis thus has to robustly segment fibers of different types and extract essential parameters with an accuracy comparable to the human expert. Currently, there are various approaches focusing on single fibers [15], networks [16–18], or bulk statistics [19]. Some of these require manual input like cell masks [20], some are tailored to certain microscopy techniques [12, 21]. However, neither of those methods provides a complete description of the whole cell cytoskeleton with fiber tracking over time. Also, most do not include a GUI and thus are not readily usable for scientist without programming experience.

Here, we present a widely enhanced version of the FS [22] with extended functionalities and unmatched usability. This ImageJ based open source software toolbox allows for robust segmentation of straight and curved fibers even in noisy imaging backgrounds that is a common concern in long-term life cell imaging. It can extract a wide range of descriptors for cellular filaments (i.e. position, length, width, angular orientation, etc.) and in particular can track and follow individual fibers over time. This single filament frame-to-frame tracking is essential to gain insights into the dynamics of both the single fiber but also interplay of all players in the cytoskeletal network. For a better overview, these new features are compiled in Table 1 and compared to the previous FS version.

This is conjoined in a graphical user interface (GUI) that also offers internal contrast and brightness adjustment, stack view, saveable filter lists and settings, on-click removable fibers, a bounding box feature to track cell events, and a new modular and object-oriented software structure under a GNU public license.

## Goals and concept of the FilamentSensor 2.0

While many scripts and packages exist and are publicly available for different approaches on fiber tracking, these are often not easily accessible for scientists without programming experience [15–19]. Therefore, we aim to provide an easy to understand and ready-to-use tool to detect and segment filamentous structures and track them over time. While the motivation in our case came from analyzing straight, contractile stress fibers in adherent cells, the FS2.0 will

**Table 1. Comparison of the feature output of the FS [22] and the novel FS2.0.** Features are exported in .csv files by the fiber detection, cell contour feature, and Single Fiber Tracking feature.

| Features | FS (0.2.2j) | FS2.0 (Alpha6.0) |
|---|---|---|
| **Cell** | | |
| Cell x-/y-position | ✓ | ✓ |
| Cell area | ✓ | ✓ |
| Cell mechanically active area | ✗ | ✓ |
| Cell long half axis | ✓ | ✓ |
| Cell short half axis | ✓ | ✓ |
| Cell contour | ✗ | ✓ |
| Cell aspect ratio | ✓ | ✓ |
| Cell mean brightness | ✓ | ✓ |
| Cell division event | ✗ | ✓ |
| Cell touch event | ✗ | ✓ |
| Cell lost event | ✗ | ✓ |
| **Fiber** | | |
| Total fiber number | ✓ | ✓ |
| Mean total fiber length | ✓ | ✓ |
| Mean total fiber width | ✓ | ✓ |
| Mean total fiber curvature | ✗ | ✓ |
| Mean total fiber angle | ✗ | ✓ |
| Total fiber order parameter | ✓ | ✓ |
| Total fiber new order parameter | ✓ | ✓ |
| Total fiber orientation | ✓ | ✓ |
| Single fiber statistics | ✓ | ✓ |
| **Single Filament Tracking data over time** | | |
| Single filament birth | ✗ | ✓ |
| Single filament death | ✗ | ✓ |
| Single filament persistence | ✗ | ✓ |
| Single filament center x-/y-position | ✗ | ✓ |
| Single filament length | ✗ | ✓ |
| Single filament angle | ✗ | ✓ |
| Single filament width | ✗ | ✓ |
| Single filament curvature | ✗ | ✓ |
| **Fiber networks** | | |
| Orientation field mean orientation | ✓ | ✓ |
| Orientation field area | ✓ | ✓ |
| Orientation field x-/y-position | ✓ | ✓ |
| Orientation field long half axis | ✓ | ✓ |
| Orientation field short half axis | ✓ | ✓ |
| Orientation field alignment | ✓ | ✓ |
| Orientation field number of filaments | ✓ | ✓ |
| Orientation field mass | ✓ | ✓ |
| Orientation field single fiber statistics | ✓ | ✓ |

be useful to investigate a wide range of questions by now being capable of analyzing all filamentous structures in the cell be they straight or curved. The fiber detection is originally based on an adapted fingerprint detection algorithm [2, 23] and embedded into a GUI that offers many additional features for image pre-processing, data handling, and export. Some of these

include cell shape descriptor extraction where area detection is enabled (area, aspect ratio, position, long/short axis) which can be combined with the fiber network descriptors, variable filter list, import function for self-written filters and area thresholding tools, stack view and handling, and on-click fiber marking and exclusion.

## Results

### Curved fibers

In its original version the FS (version 0.1.7) [22] could only detect straight filaments. However, cytoskeletal structures in cells, can also be curved. While this is well known for microtubules and intermediate filaments, also acto-myosin stress fibers, sometimes exhibit curved structures either due to internal network topography (e.g. connections to force chains [24] or external cues [25]). We therefore implemented a step-wise forward searching algorithm to map curved filaments. An intermediately updated version of the FS (version 0.2.3, see [26]) was able to detect curved filaments with limited customization ability. Now, the FS2.0 comes with a variety of adaptable parameters to fully utilize the possibilities of customization and automation for reliable detection of cytoskeletal structures in cells.

A common approach to detect curved filaments is to use segmentation and stitching [12] of filaments. Here, we use a forward search starting from seeds as illustrated in Fig 1. Those seeds are elongated with segments of variable length and curvature and adjustable boundary conditions to identify different curvatures and filament lengths of filamentous cytoskeletal structures while reducing false positive detection of filaments introduced by noise or preprocessing.

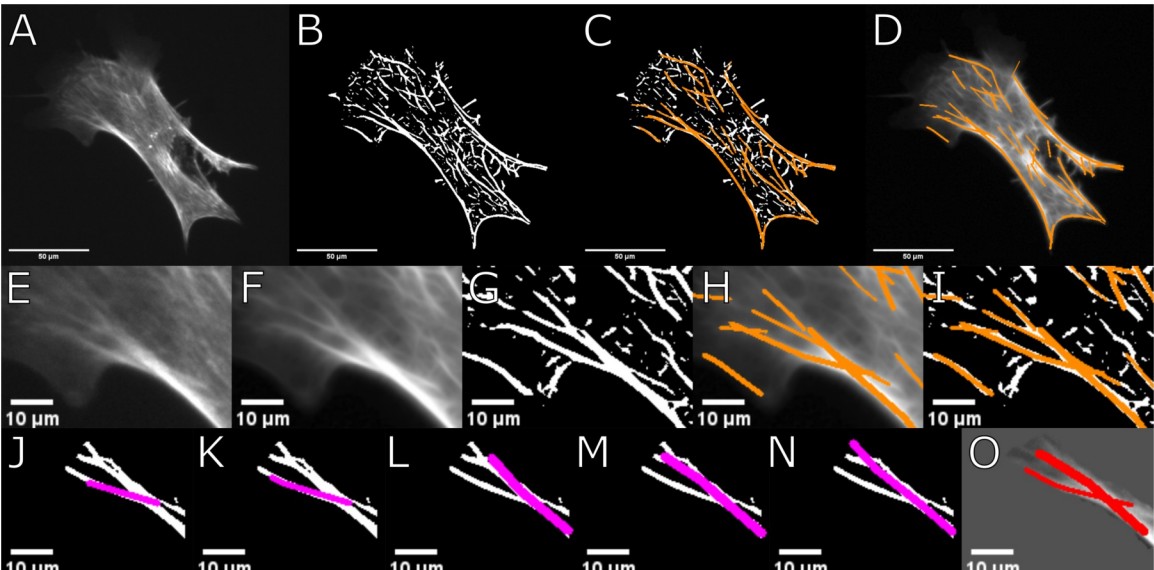

**Fig 1. CurveTracer class step-wise illustration. A**: Raw image of example cell. **B**: Binarized image of example cell. **C**: Filaments recognized by FS2.0 shown in binarized image. **D**: Found filaments shown as overlay of raw image. **E**: Cropped example region of cell shown in A. **F**: Output of filter routine preparing raw image for filament detection. **G**: Binarized image of example region. **H**: Altered image and found filaments overlayed in orange. **I**: Found filaments in binarized image and overlayed in orange. **J-N**: Visualised examples of different potential filaments found by curve tracer glass gather function. **O**: Final filaments found by curve tracer class after correction and weighting of all possible filaments generated.

In the FS2.0, after the original image is pre-processed and binarized, the `CurveTracer` class is utilized. This uses the binarized image to probe the neighborhood of each white pixel at an angle to find the largest diameter of white pixels. This gives sequences of straight pieces with maximal diameter of white pixels, resulting in a matrix of pixels and width values.

The `CurveTracer` class appends the `LineSensor` class used to find the largest diameter of white pixels on a straight line. Linking multiple straight lines to a curved one might seem insufficient but with the option to adjust the value for minimally required straight pieces $\ell_{str}$ and the minimum angle difference $\phi_{diff}$ and tolerance angle $\alpha_{tol}$ one can adjust the algorithm to find highly curved fibers. Input parameters for probing are defined as:

- Minimum filament length $\ell_{min}$: minimal length of total curved filaments.

- Length of straight pieces $\ell_{str}$: maximum number of new found filament segment added to the existing filament.

- Minimum angle difference $\phi_{diff}$: factor to vary the hard-coded 3˚ probing increments.

- Tolerance angle $\alpha_{tol}$: default is 20˚ and sets a threshold for 'conflicting orientation'.

The found curved filaments consists of linear pieces. For each white pixel, the `CurveTracer` probes into all directions with 3˚ increments with twice the length of minimal straight pieces, $2 \cdot \ell_{str}$. The average width value is determine in every direction and the two opposite directions with the largest average width are selected.

Then, endpoints of the line are moved by $\ell_{str}$ into these directions. If the `linewidth` in this direction is not sufficient (below 1), pixels from the end of the line are removed up to the point with sufficient `linewidth` (above 1) and a new endpoint is set there. From each endpoint, the average width is calculated for the direction of the last step and neighboring directions and the direction with the highest average width is chosen for the next step. This is repeated iteratively.

When the `CurveTracer` searches are finished for both directions, the filament is stored if the minimum filament length $\ell_{min}$ is exceeded. In the next step the orientation field is used to double check the found filaments. All stored filaments are called, longest filaments first, and for each filament the longest segment first. For each segment, the orientation is determined for each pixel. The segment is only accepted as valid if less than 30% of pixels have a conflicting orientation. The limit of this conflicting orientation is set by the tolerance $\alpha_{tol}$ (default tolerance angle 20˚). This orientation and the segments are then stored.

In case of more than 30% conflicting orientations, the segment is either discarded or shortened. Discarding occurs if the endpoint orientations do not carry a conflicting orientation. Shortening will be done from the conflicting endpoints inwards till a point without conflicting orientation is reached. If the remaining filament length is above $\ell_{min}$, it will be stored again and revised when it's new, shorter length is called.

The detection of curved fibers can be applied to all fibrous elements of the cell as demonstrated in Fig 2 for microtubules, vimentin, and actin filaments in hMSCs. For this, the three customized, re-importable settings used can be found in the DOI (https://doi.org/10.5281/zenodo.7314056). Those differ mainly by the required minimal length of filament pieces, the forward search, and maximally allowed angle between filament pieces. The settings can be adjusted to each type of filament type to reduce false-positives. A summary of this analysis is shown in Fig 2 displaying filament length and curvature, that vary with respect to filament type and substrate stiffness.

This example shows one kind of bulk analysis with the novel FS2.0 that can give valuable insights into the cytoskeletal composition.

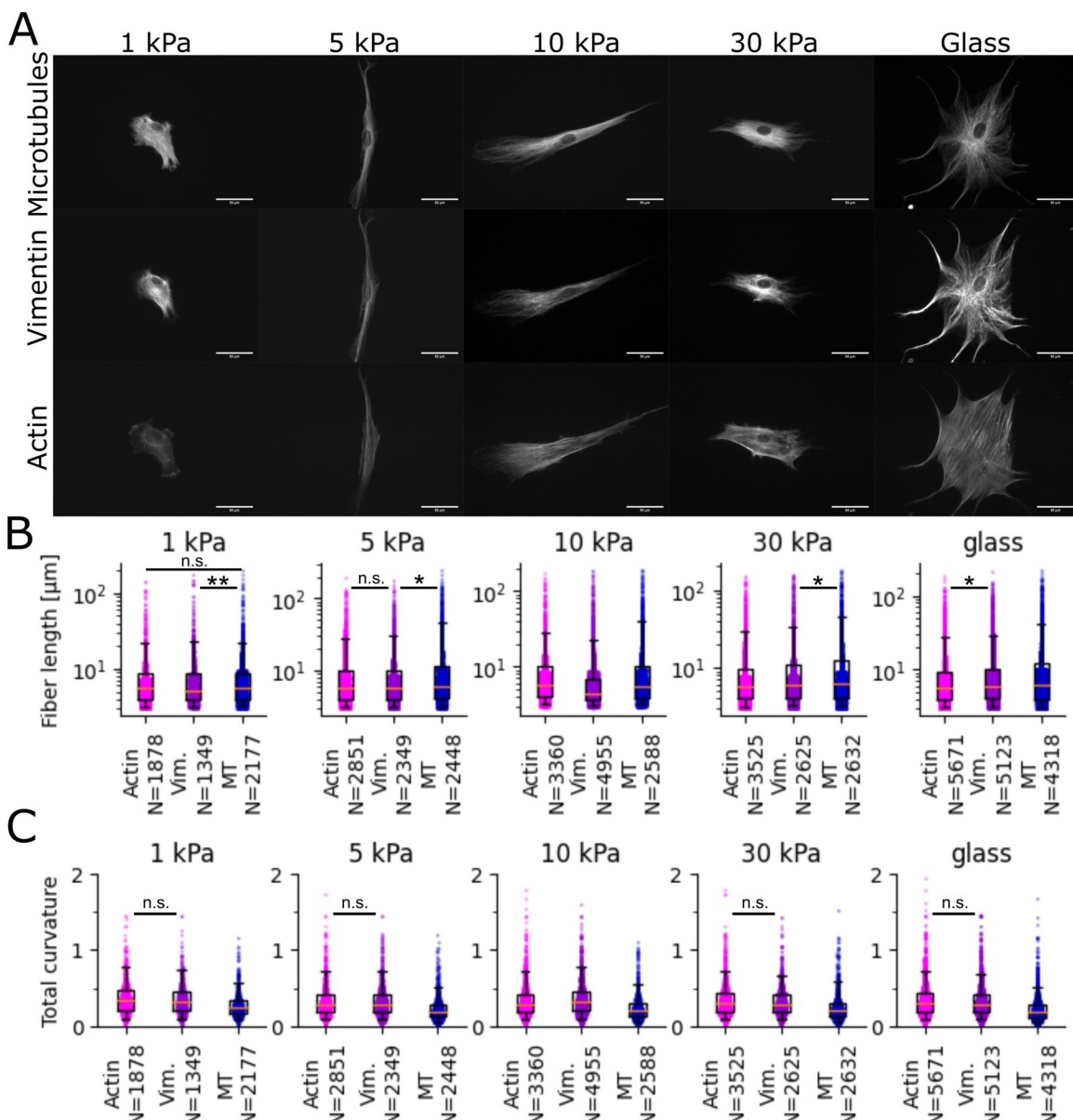

**Fig 2. Example data set with different cytoskeletal fiber types. A**: Exemplary pictures of the data set. For every stiffness (1,5,10,30 kPa PA gel, and glass) 30 images of 5 channels (phase, tubulin, vimentin, actin, DAPI) are available. **B**: Comparison of filament length found for microtubuli, vimentin, and actin using FS2.0 with settings optimized. **C**: Comparison of filament curvature found for microtubuli, vimentin, and actin using FS2.0 with settings optimized. We performed a Kolmogorov–Smirnov test between each pair of conditions. n.s. = >0.05, * = <0.05, ** = <0.01, all other conditions <0.001.

## Comparison using ground truth data

For benchmarking purposes we recorded an improved ground truth dataset for this paper. It consists of more than 300 fixed cells per condition stained for stress fibers and nucleus. We seeded precultured hMSCs on elastic polyacrylamide (PA) gels with different Young's moduli ($E$ = 1; 2; 5; 11; 20; 32 kPa) and glass all coated with collagen-I. Preparation was done as in our earlier work [7, 8]. Cells were fixed after 24 hours and stained for actin (Phalloidin-Atto550) and DNA (Hoechst) with three replicates per condition (except 2 kPa, there $n$ = 2), to visualize

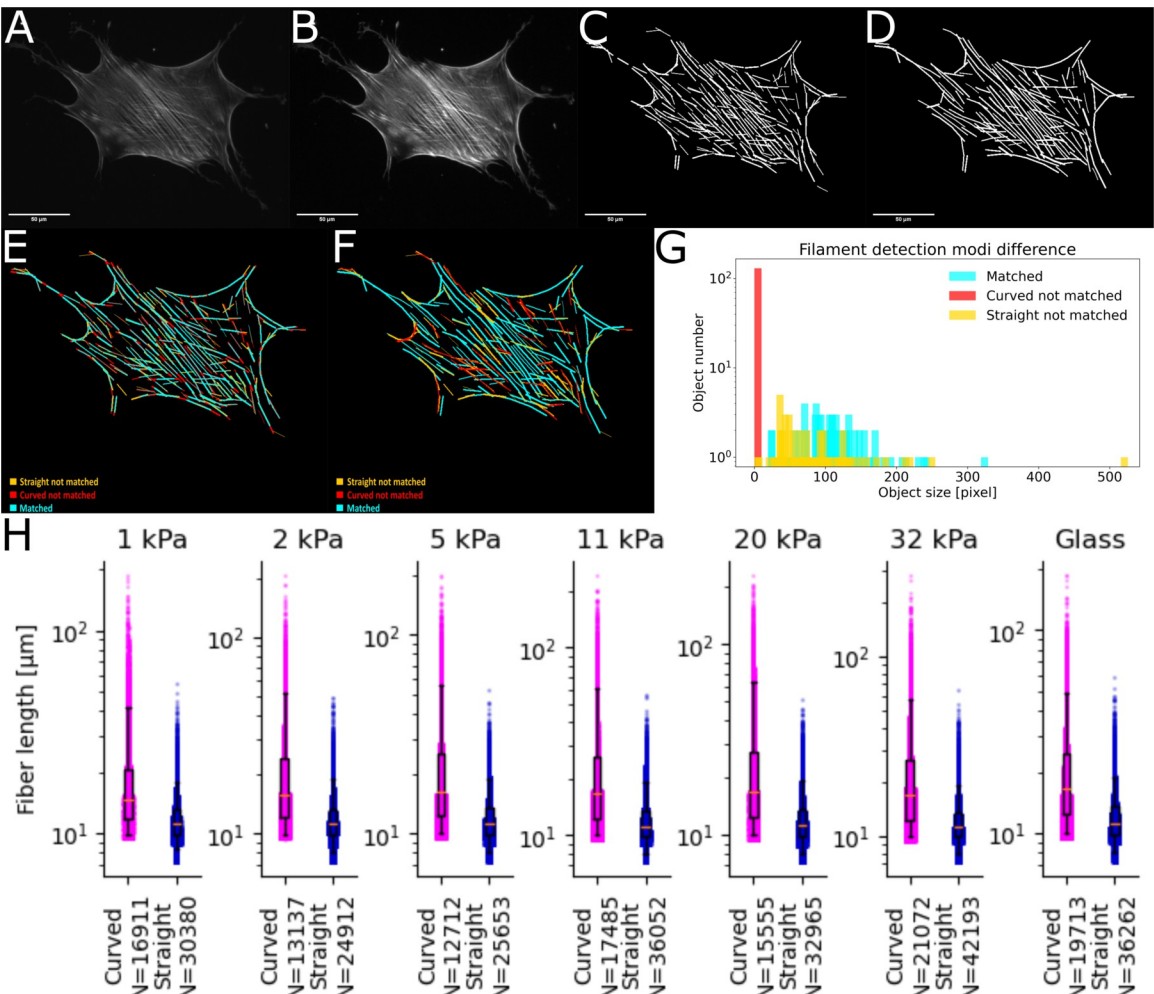

**Fig 3. Example of straight versus curved filament tracing. A**: Raw image of hMSC on collagen-I coated glass. **B**: Result of FS2.0 filtering during pre-processing. **C**: Filaments found using straight filament detection. **D**: Filaments found using curved filament detection. **E**: Pixel-wise comparison of filaments found with straight and curved settings. Straight settings used as ground truth. Orange: Pixel only in straight detection. Red: Pixel only in curved detection. Blue: Pixel present in both settings. **F**: Object-wise comparison of filaments found with straight and curved settings. Straight settings used as ground truth. Orange: Object only in straight detection. Red: Object only in curved detection. Blue: Object present in both settings. Comparison was done with setting so that 75% of the object needs to be matched to count as same object. **G**: Histogram of object size for all three filament categories, colors matched to pictures. **H**: Comparison of all filaments in the ground truth data set for straight and curved settings. We performed a Kolmogorov–Smirnov test between each pair of conditions. All p-values are <0.001.

stress fibers and nuclei, respectively. For this ground truth data set, only isolated cells were imaged using an inverted fluorescence microscope (Zeiss CellObserver, 20x objective) and a sCMOS camera (Andor, Zyla). The respective raw images as well as the analysis and the used parameter files (settings) for straight and curved filament detection can be found in the DOI (https://doi.org/10.5281/zenodo.7314056).

Fig 3 summarizes the differences of straight versus curved filament detection for an example cell on glass as well as the bulk differences. Panel 3 H shows that detection restricted to straight fibers finds double the amount of single fibers and detected fibers are significantly shorter. Thus, curved fiber detection fundamentally improves detection and analysis, while preventing artificial partition of fibers and conserving the true structure.

## Applicability

The FS2.0 is optimized for 2D images and stacks thereof applicable to a wide range of fluorescence microscopy image qualities. Our ground truth data shown in Fig 3 as well as the multi-fiber data shown in Fig 2 show wide-field epi-fluorescence microscopy images, a technique available in most laboratories. However, the FS2.0 can be also applied to super-resolution microscopy data as well as images of fibrous networks. The FS2.0 repository contains example data of membranous f-actin and test routines with parameters optimized for network analysis. This was possible by refactoring code which relied on percentage of cell area per image. Thus, network images can be analyzed in the FS2.0 and cell descriptive features in the data output ignored. Table 1 shows an overview of the standard feature output of the FS2.0 and FS [22] for comparison. These features are exported in .csv files by the main fiber analysis, the cell contour feature, and Single Filament Tracking feature. Example data can be found in the DOI. A designated 3D application of the FS2.0 is not yet available. However, it is possible to use and tailor the Single Fiber Tracking routine to stitch fibers in *z* instead of over time.

## Single fiber tracking

One of the new key features of the FS2.0 is to follow individual fibers in space over time, namely single fiber frame-to-frame tracking. Following the structural evolution and dynamics of single filaments over time is key to understand the fundamental mechanisms of the cellular cytoskeleton to elucidate the complex mechanical interplay of cells with their surroundings. Here, a major aim is the classification and analysis of filament structure and turnover by objective and unbiased parameters that allow for quantification and subsequent modelling. During certain cell events, different cytoskeletal components and sub-types are involved and remodeled. Thus, identification of relevant fiber sub-populations as well as further in-depth analysis of the dynamics will open up the field for future research.

Fiber tracking naturally presents as a transport problem, as described by [27, 28], a field of research which has received increasing attention in past decades, see e.g. [29]. The method applied here for fiber matching is based on the Wasserstein distance described first by Vaserstein [30]. Since in the case of a dynamically changing cytoskeleton number and length of fibers can vary over time we use a Wasserstein distance for unnormalized measures, based on the approach by [31]. As a base distance we use a combination of maximal distance between fragments, angle difference for transport, line distance for transport, and used fragment length detailed below. Using this base distance, the Wasserstein method takes into consideration angle shifts, thus it is well suited for moving cells where the cytoskeleton becomes laterally distorted over time.

Single Filament Tracking can be executed any time after filaments have been extracted. The FS2.0 uses the Wasserstein distance to find corresponding filaments from consecutive frames and link their identifiers to gather a new object with parameters for each time point. The data can be exported either grouped by object or by time point. To offer easy visual control, we show filament lifetimes and cell overlay in the GUI (see S1 Fig in S1 File). These are furthermore coded for their individual persistence over time, supporting the visual quality control. The single filament data itself contain all filament descriptors in a similar format as before.

In Fig 4 we show the results of single filament tracking for an example cell in a 24 hour movie. For a selected time point (15 hours after seeding) we show raw image (Fig 4A), tracked fibers that are present in more than four consecutive time frames (Fig 4B), and fibers only detected in one frame, considered noise (Fig 4C). Additional, we show bulk development of detected filaments over 24 hours after initial seeding (Fig 4D) and detected noise over the

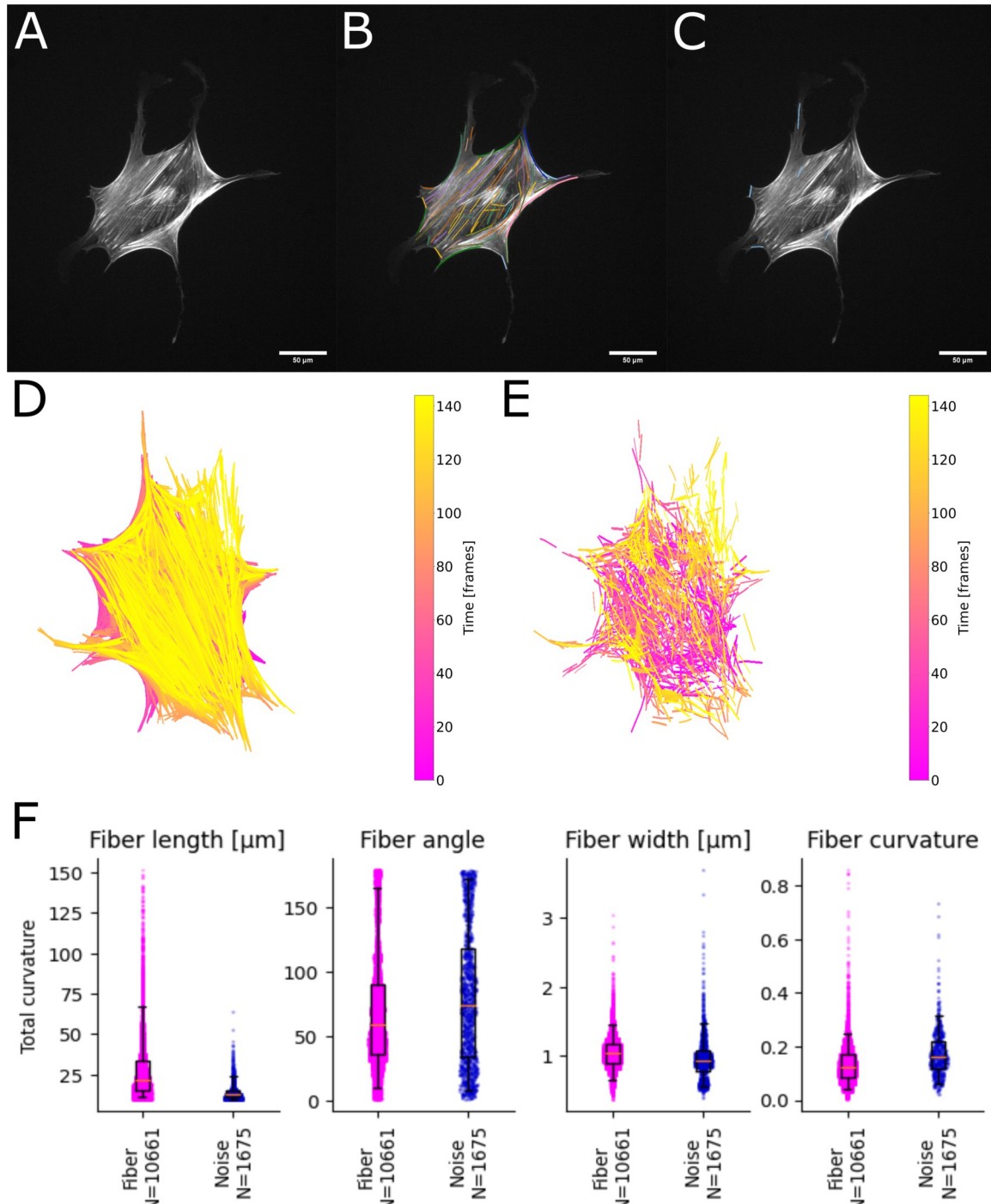

**Fig 4. Single filament tracking. A**: Original image of hMSC on collagen-I coated glass, 15 hours after seeding. **B**: Filaments found in the example frame with lifetimes longer then 4 frames randomly color-coded. **C**: Filaments found with the single filament tracking plugin in the example frame with lifetimes of only one frame. **D**: Overlay of all filaments with lifetimes larger then 4 frames color-coded for frames. **E**: Overlay of the noise with lifetime of one frame color-coded for frames. **F-I**: Comparison of trusted filaments versus noise for: **F**: length, **G**: width, **H**: angle, and **I**: curvature. We performed a Kolmogorov–Smirnov test between each pair of conditions. All p-values are <0.001.

same time (Fig 4E). This analysis shows that noise is detected across the whole cell and more noise is detected at earlier times, consistent with progressing cytoskeletal structure formation.

## Bounding box for cell event detection

As in unsupervised live-cell imaging large data sets can be generated, a tool to automatically identify cell events can drastically reduce time spent on analysis.

In long-term single-cell imaging certain events usually lead to exclusion of the respective cell from analysis. These include in particular, division, apoptosis, cell-cell contacts, or cell-debris contact. In the FS2.0, we implemented a bounding box and area tracker to follow cells from frame to frame and compare their respective areas to deduce cell events. Additionally, this feature can be used to select one cell for analysis in frames of multiple cells.

The bounding box is generated from the utmost top, left, bottom, and right point of the cell area. We chose to use a rectangle as it is fastest performance-wise. Using the area outline directly is ill advice due to its strong dependence from the quality of area outline detection. A polygon of the outline would be both fast and sufficiently accurate.

From the bounding box, a 'matching map' is derived. Each area detected by the `CellShapePlugin`—according to the constraints set there—for each frame in a stack will be matched to the neighboring frames using the bounding box as search constraint. A matching score is calculated by the relation of the overlap between both area to the smaller of both areas. If this is larger than unity, the score is accepted and saved in the 'matching map' with both matching areas features.

Subsequently, it will be iterated over every frame in the stack. For the frames where an area is detected first, the 'dynamic area' is initialised and an 'appearing' cell event is generated meaning that a cell is detected for the first time. Of course, this also will happen for dirt of sufficient size like dead floating cells. For our purpose, this lacking discrimination is of no relevance, as floating cells can interfere with the structure detection as well. If the cell is found in more than two frames, the areas for past frames are retrieved from the matching map together with the matching score, if this score is above the intersecting tolerance. The intersecting tolerance is a parameter that the user can set. The scores are then grouped to:

- Cells with same predecessor: `CellSplitEvent`

- Cells without predecessor: `CellStartEvent`

- Cells with precisely one predecessor: `CellAliveEvent`

- Cells with more then one predecessor: `CellFusionEvent`

For each individual confirmed cell object a new `DynamicArea` 'lifeline' is created and all linked area and cell events with respective frame number are added. Matching `DynamicArea` 'lifelines' are reduced to one. Fusion and split events hereby start a new 'lifeline'.

After all frames are processed, a post-processing step validates the 'lifelines'. Here, `CellFusionEvents` is iterated whether later a `CellSplitEvent` occurs that would be a `CellTouchEvent` with `CellDeTouchEvent`. If that is the case, lifelines are corrected. To assign the correct previous cells, the most similar matching score will be used. However, here errors can occur. First, if both cells sizes change drastically. Second, if the cell touching occurs at the end of the stack and no de-touching is registered, it will be counted as `CellFusionEvent`. At last, for all cells without successor, a `CellEndEvent` is noted. Then, variables 'birth' and 'death' are set for each `DynamicArea`. Here, again, the `CellSplitEvent` and `CellFusionEvent` cause 'birth' and 'death' in the 'lifelines' while touching and

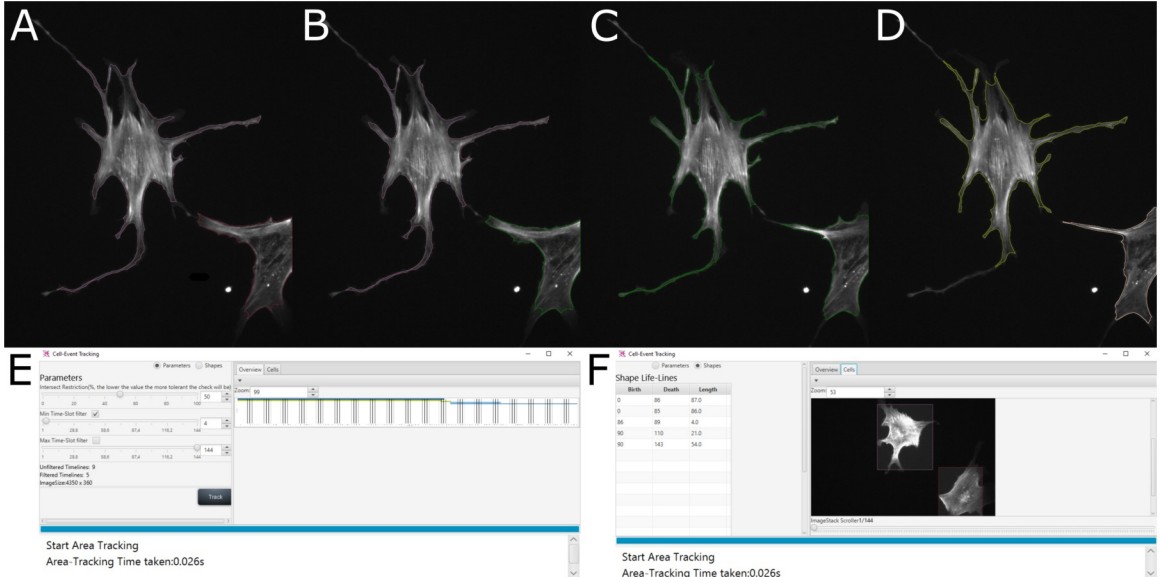

**Fig 5. Bounding box feature for cell event detection.** Example visualization of cell events as they appear in the GUI: **A**: Initial images, two cells recognized. **B**: Color of right cell changed due to edge touching. **C**: Cell contact detected based on area outline of both cells, cells appear in same color. Note that the shown area outline does not stretch to the cell extension that is responsible for the 'contact' flag, as the area output of the FS2.0 is more filtered than the stage used for the bounding box feature. **D**: Cells de-touched again, both cells are assigned new lifelines and new colors. **E**: Bounding box feature window with filter options and lifelines found visible. **F**: Bounding box feature window with marked cells and cell lifeline information visible.

detaching will not interrupt the lifeline. Again, if no de-touching event for cells can be registered, the 'death' will be assigned falsely.

To visualize the detected 'lifelines' for the user for easy inspection and adjustments, the bounding box feature appears as separate GUI in the FS2.0 (shown in Fig 5). The user can choose between 'lifeline' view (see Fig 5E) or cell boundary and event view (see panel Fig 5F). Images Fig 5A–5D show an exemplary progress of cells touching and de-touching.

## Usability

Besides stack handling, brightness and contrast adjustment, other functionalities have been added to make the analysis with the FS2.0 a pleasant experience. Most noteworthy are the saveable and importable filter list and settings file encompassing all possible settings in preprocessing and filament tracking. This offers the possibility to customize settings to fiber type, image quality and ensures a reproducible analysis.

Importantly, these lists and settings can not only be re-imported during processing of a single file but also in the batch processing mode. The batch mode is designed to allow for automatized processing of an arbitrary number of sub-folders containing single or stack images. This is especially useful for series of experiments or high-throughput microscopy and provides an image analysis pipeline that can keep up with the generated large data sets. Resulting files will be stored within these sub-folders and can be analyzed easily due to their serialized generation. Additionally, other features like single fiber tracking, orientation field mapping, or outline detection can be included into the batch mode.

As Fiji already includes most routines for image analysis, we adapted their routines and functionalities wherever possible. Nonetheless, some users might want to write own filters for area detection or smoothing. To enable this, we introduced interface classes. Thus, the user

can write their filter according to a template provided and easily integrate it into the existing software to enhance functionality.

## Refactoring and runtime improvement

The total runtime of an algorithm is a crucial parameter when considering real-time or near-real-time analysis. This is in particular of interest for memory intensive applications like time-lapse life cell microscopy.

The FS2.0 has undergone major refactoring to improve inter-usability. We have implemented structuring in classes, serialization, interfaces, and multi-threading. Aside from a significant reduction in runtime, the resulting structure and code is well-documented and easy to re-use. Additionally, we included test routines and benchmark data. Selected core classes are and dependencies are shown in Fig 6 and the whole class tree with dependencies can be found in the DOI as well as in the repository itself.

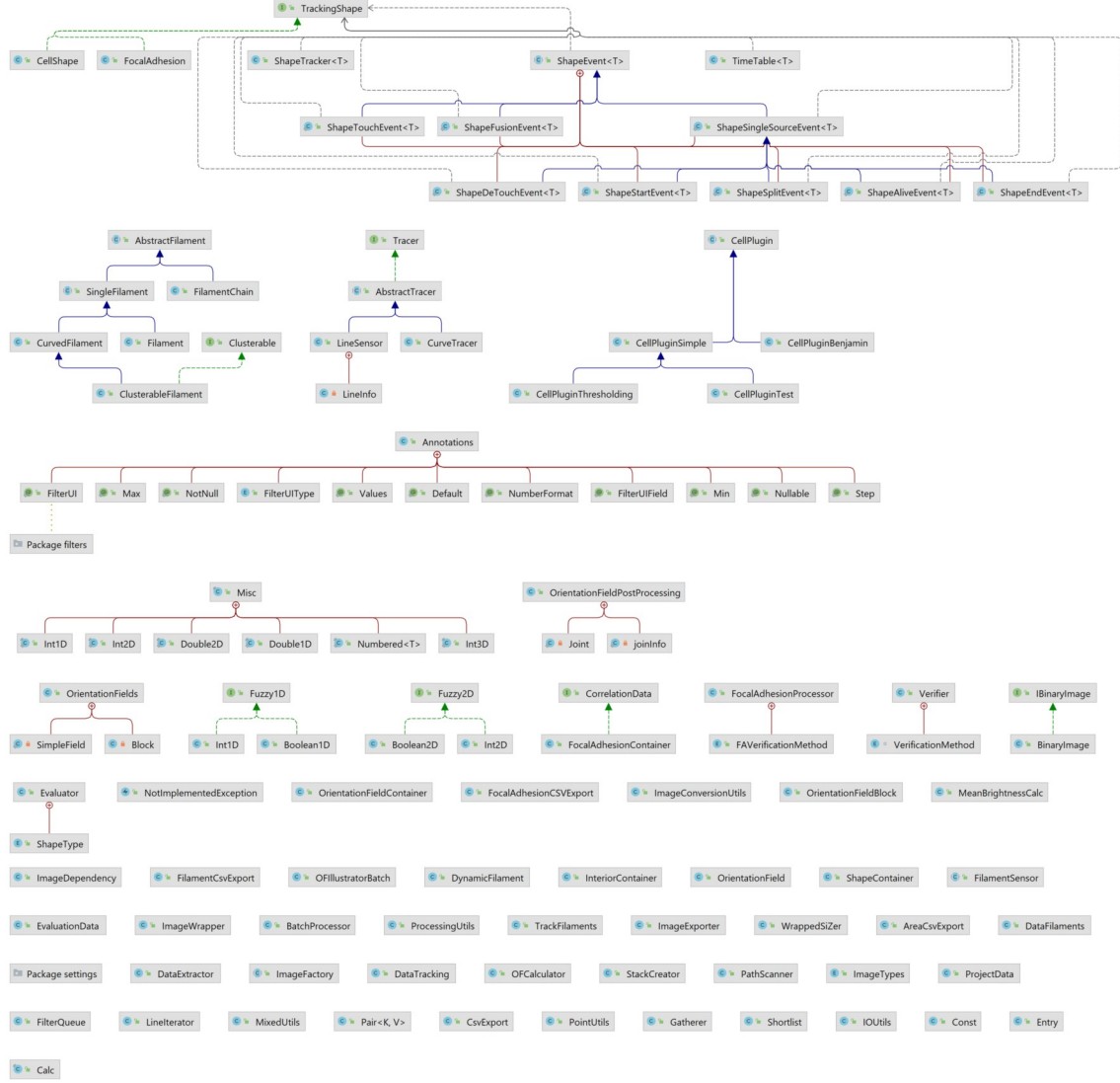

**Fig 6. Class tree of the main FS sources.** Reduced class tree of classes necessary for fiber tracking and annotation. Wherever possible dependencies are avoided, thus main classes like `DynamicFilament`, which is used for single filament tracking, can be reused easily.

**Table 2. Comparison of runtime of different FS versions before and after refactoring.** Version 0.2.2j was published by Benjamin Eltzner [22], version 0.2.3 is the first version after start of refactoring process. Version Alpha3.0 is an unpublished developer version with refactoring process completed. Version Alpha6.0 is using `OpenJDK15`.

| FS 0.2.2j JDK8 | FS 0.2.3 JDK8 | Alpha3.0 JDK8 | Alpha6.0 OpenJDK15 |
|---|---|---|---|
| 1324 s | 1593 s | 219 s | 125 s |
| 1322 s | 1561 s | 215 s | 124 s |
| 1321 s | 1575 s | 211 s | 125 s |
| 1322 s | 1576 s | 215 s | 125 s |

After refactoring, the processing time was reduced by a factor of ten. From the FS version 0.2.2 [22], which took 1323 s to process our standard test data set of 144 images, each with dimensions of 1560 x 1866 pixels, gray scale (S3 Fig in S1 File), runtime increased to 1593 s for the FS version 0.2.3 [26]. With the first round of refactoring we could reduce the runtime to 218 s for FS Alpha3.0 and with further polishing almost cut it in half again to 125 s in the FS Alpha6.0 utilizing OpenJDK15 and OpenJFX. We tested different versions of the software on a system with the following specifications: Windows 10 64bit; Java: java version "1.8.0_172" or "OpenJDK15" 64bit; Processor: Ryzen 2700X (8 Core 16 Threads, 3.7GHz—4.3GHz); hard disk–data was written to: USB 3.0 external hard disk; RAM: 32GB @2933Hz (dual-channel); NVME SSD: read/write >2000MB/s. Time is measured from `Batch Processing` from start of `Java main` until end of `main` method. Timestamps are taken via `System.currentTimeMillis()`. Each time a cold start is done (one run of batch processing, then end). Test data is the same for every version of FS. For comparison, three runs of the same version are done and averaged. The FS versions are modified by adding a method-call to start batch processing from an external program.

As Oracle increasingly reduced open usage of the `JavaJDK` from version 9 onward, we decided to switch to `OpenJDK15` for the FS2.0, which should remain open source indefinitely. All stable versions of the FS, published and developer versions, are available online at www.filament-sensor.de.

With significantly reduced runtime, real-time analysis as a plugin for MicroManager [32] and PycroMangager [33] can now be developed. Furthermore, storage-intensive tasks like stack handling became possible. Fiji [34] is widely used, and users are accustomed to easy screening and editing of files in stack view. With better RAM usage and multi-threading, stack handling and multiple editing options applicable to either single frames or the whole stack are the center of the improved GUI, offering the user fast visual control of the analysis process. Additionally, the FS2.0 now supports the same range of file types as Fiji Table 2.

On average, version Alpha6.0 needs only 8% of runtime in comparison to the original FS [22] and is using available CPU and RAM more efficiently. This was an essential requirement for implementation of more runtime intensive operations as stack processing and in particular single filament tracking.

- FS Vers. 0.2.2j: CPU usage 0–10%, memory usage up to 3 GB, most of the time below 2GB.

- FS Vers. 0.2.3: CPU usage 0–11%, memory usage up to 3 GB. No performance difference when compared to Version 0.2.2j.

- Alpha3.0 and Alpha6.0: CPU usage 0–100%, memory usage up to 6 GB. Note that the process exited about 59 seconds after batch processing finished, this is because the Java threads from `Stream API` and `CompletableFuture.runAsync` are shutting down. This has no practical impact on performance.

We advise using FS2.0 with `OpenFX` and `OpenJDK15` as those are open source packages and offer best resource usage.

## Methods

### Curved fibers

With the restriction to straight lines, the original FS was not able to account for curved fiber types like microtubules, intermediate filaments or bent stress fibers. Thus, in version 2.0 the `CurveTracer` class appends the `LineSensor` class.

The `LineSensor` class contains the `scanFilaments` method that calls the `calc-WithMap()` method. This `calcWidthMap()` iterates through diameters of the `width-Map` matrix. The `widhtMap` is created by overlapping each pixel with a circle mask and the amount of misses (pixel in binary image not white) to hits (pixel in binary image is white) is are calculated. This amount of hits to misses that needs to be met is influenced by the tolerance setting. If the condition is met, the circle diameter is increased till it is not met anymore. The maximum diameter is inserted in the `widthMap[x][y]` cell in the matrix.

For the maximum diameter in this `widthMap` and for all diameters a binary image is created where `width_map[x][y]` > (currentDiameter-1), with only pixels with larger diameters than the current. The condition is negated since the sensor interprets false as white pixels.

Afterwards the `findOrientations()` method is called and the `LineSensor` initialised. For every white pixel `SpokeLattices` and `LineInfo` are used where `LineInfo` is the class used for line identification. This created a map of all points on lines whose length exceeds `minimal_length`. The points are sorted in lists by line length. Then all remaining lines are run through (longest first) and marked in `m_orientation_field`. The resulting filament is added to `filament-list`.

The `CurveTraver` class uses the `initSpokeLattice()`, extending the one from the `LineSensor` class. Using the first part as described but using different, not constant values, the second part then creates two lattices with different depth's.

The `getWidthMap()` method then returns a calculated `widthMap` (extending `calc-WidthMap()` method) which groups all points with the same diameter into a list `(Map<Integer,List<Point>>)`. As in the `LineSensor`, starting with maximum diameter, it is iterated through all diameters in the `widthMap`.

For all points with the current diameter `scanFilamentsInitScores()` method initializes scores and objects of the `gatherer.sense()` class that does the curve sensing. Starting at `m_start` points meeting conditions are sensed and a score is returned which is the sum of all width's from `width_map` per point. The `getBestPair()` method then checks opposite directions and direction change in the start point and gets best pair of gatherers based on the score. After that the gather method `scanFilamentsGetGathererLine()` is called and the gatherers are joined and returned. Here, `gather()` gathers points on curve and adds points to `m_points` and `join()` joins two gatherers into one (sum points and scores of both gatherers). The fused gatherer is added to a `scored-line-map` and the `max_score` is updated if the new gatherer's score is higher than current `max_score`. If a certain `lineCounter` is exceeded, the `cleanUpLines()` method is called. This `cleanUpLines()` runs through the longest lines and marks them and re-sets `max_score`.

After that the `markLines()` method is called. It runs through lines (longest first) and marks them. Inside `markLines()` the method `drawLine` is used, which adds the filament to the `filament-list` and updates the `orientationField` list.

## Single filament tracking

The FS2.0 offers a fast way to visualise and export fibers tracked over time. Furthermore, the user can set 4 out of the 8 parameters fundamental to the Wasserstein transport plan and thus further adjust the tracking to their type experiments. Regarding the hard-coded parameters, they consist of:

- The minimal percentage of a filament, that has to be matched to future filaments to be valid

- The length of a filament, that has to be matched

- Relative size of the filament to sort filaments of the next time step that can be matched

- Absolute minimal size of filaments of the next time step that are considered for matching

As these parameters are mainly used for noise handling and reduction. Thus, we do not advise to change them unless time for testing can be invested. They are, however, easy to find in the source code. Of more relevance for the basic user is the change in the cost function parameters that are incorporated in the "Single Filament Tracking" pop-up window. Here, the all relevant parts of the cost function can be set:

- Maximal distance between two fragments for matching

- Factor for the angle difference in the Wasserstein transport

- Factor for the line distance in the Wasserstein transport

- Used fragment length where shorter fragments rise both accuracy and processing time

Additionally, minimal and maximal desired persistence time of the filaments can be set. The output of single filaments over time is given both in `csv` and `png` files. For `csv` output, user can select between data bundled by frame (time point) or filament identifier. In the visual output, fibers are color-coded for their persistence length and thus easily distinguishable.

To track stress fibers of cells in life cell movies over time we need to match filaments from one frame to the next. Here, we use the concept of 'optimal transport' to solve this task [27]. In the classical formulation one considers a set of producers $P = \{p_j\}_j$ which supply various volumes of a particular commodity and a set of consumers $C = \{c_k\}_k$ with varying demand. The question is now, how to optimally use trucks to move the commodities from producers to consumers such that the demand is met. For this, one has to know the individual transport cost $d_{jk}$ for each pair of producer and consumer. One can then optimize the deployment of trucks to minimize the total transport cost $T$ as elaborated in more detail in the supplement.

In our case, fibers are cut into short pieces and these pieces are the producers in the first image while pieces in the second image are treated as the consumers. To create a map between fiber pieces, we are not interested in the transport cost $T$, but in the transport plan, i.e. the truck routes. For each fiber piece in the first image, we determine the fiber pieces in the second image to which a transport occurs. Furthermore, we allow for fiber pieces to remain partially unmatched, if no nearby transports are possible. For the transport cost we not only take into account the distance between pieces but also their relative orientation in order to achieve meaningful results.

The resulting correlation matrix between fiber pieces on both images can then be consolidated to a correlation matrix between the full fibers by simply taking the sums over blocks in the correlation matrix.

**Wasserstein distance.** Assume two sets of objects, producers $P = \{p_j\}_j$ and consumers $C = \{c_k\}_k$, a *base distance* or *cost function* $d : P \times C \to \mathbb{R}_{\geq 0}$ and *weights* for every element of $P$ and

$C$, which we denote by $w(p_j)$ and $w(c_k)$ respectively. Define an *optimal transport* as a matrix $T = (t_{jk})_{jk}$ satisfying the following minimization with constraints

$$T = \operatorname{argmin} \sum_{j,k} d(p_j, c_k) t_{jk}$$

$$\forall j : \sum_k t_{jk} = w(p_j)$$

$$\forall k : \sum_j t_{jk} = w(c_k)$$

where "argmin" simply denotes the minimum position, not the minimum value. As a side remark, the transport plan does not need to be unique in general, but for many choices of $d$ it is unique in practical situations.

For the optimal transport $T$, the Wasserstein distance is $\mathcal{W}(P, C) = \sum_{j,k} d(p_j, c_k) t_{jk}$. However, we will only use the transport plan.

**Filament transport.** In our case, the sets $P$ and $C$ contain line segments of roughly 10 pixels length from two subsequent images in a movie. the weights $w(p_j)$ denote the length of each line segment in pixels. Both $P$ and $C$ contain a "dummy" element $p_0$ and $c_0$, which have distances $\forall j : d(p_j, c_0) = d_{max}^2$ and $\forall k : d(p_0, c_k) = d_{max}^2$ and "infinite weights" $w(p_0) = \Sigma_k w(c_k)$ and $w(c_0) = \Sigma_j w(p_j)$. Thus a distance of more than $d_{max}^2$ between two segments means that they will not be connected by the Wasserstein transport. In this sense the transport is "local". For all true line segments, the distance is composed out of $d_l(p_j, c_k)$ which is the distance in pixels between the closest point of the two segments and $d_\phi(p_j, c_k)$ the orientation difference between the two segments in degrees. Then the base distance is

$$d(p_j, c_k) := 4 d_l(p_j, c_k)^2 + d_\phi(p_j, c_k)^2, \tag{1}$$

which, together with the dummy elements means that only line segments which are at most $d_{max}/2$ pixels apart and whose orientation differs by at most $d_{max}$ degrees can be joined. (We typically use $d_{max} \approx 20$.)

The transport plan $t_{jk}$ is then interpreted as something similar to a correlation matrix.

**Implementation.** Usage of the Wasserstein distance in the software has been implemented to have 4 parameters adjustable in the GUI to accommodate different fiber lengths and curvatures.

Parameters are implemented as follows:

- double `max_dist`: maximal distance of fibers that may be matched

- double `factor_angle`: Pre-factor of the angle difference in the distance squared calculation

- double `factor_length`: Pre-factor of the line distance in the distance squared calculation

- int `length`: Length of the used line fragments; shorter fragments increase computation time (cubic) and accuracy

Furthermore, there are four constants in the `DataTracking` class that can be altered to improve on the Wasserstein transport:

- double `MIN_VALID`: Minimal portion of a filament that has to be mathced for the filament not to be discarded as 'solitary'

- double `MIN_MATCHED`: Fraction of the filament length that has to be matched from the next time point for the `trackFilaments` method to stop adding filament fragments

- double `MIN_REL_SIZE`: Minimal fraction of length of the current filament a filament from the following time point has to have to be matched

- double `MIN_ABS_SIZE`: Minimal length a filament of the following time point has to have to be added

**Visualization.** Single filament tracking over time relies on already segmented and analyzed fiber tracking data of all frames. Thus it can only be opened from the filament sub-menu once the fiber data is gathered or after loading a previously analyzed project with saved filament data. On click it opens a separate window where segment length and shifting angle variables for the fiber transport can be changed, as those are dependent on the data set analysed. With more motile cells and longer pauses between images, more transport and angle shifting has to be permitted. For visualisation, minimum and maximum of the desired stay time can be set. The GUI is shown in S4 Fig in S1 File.

The fiber populations are shown according to their duration of appearance next to the single filament tracking sub-menu. On click a panel can be opened to view the time stack with the found filaments marked color coded according to their appearance duration.

## Bounding box

The bounding box feature creates a rectangle that contains the whole cell and is calculated from a thresholded outline of the cell. We use a box here as with cytoskeletal staining cell structures as filopodia might be not detected. Thus, a box offers more overlap and is the fastest method in respect to runtime. However, a polygonal shape would be more accurate. The convex hull, however, is both extremely accurate and already exists as output in the area results. The bounding box can be substituted with this, if needed, with significant influence on runtime.

The bounding box uses following parameters:

- `i`—running variable

- `maxTime`—number of images in a stack (each picture a time point)

- `intersectTolerance`—adjustable parameter how much overlap is tolerated till cells count as touching

First, a 'matching map' is created. Each detected area of a picture in the stack (`i`) will be compared to detected areas in the next image (`i+1`) and a matching score is calculated. This score is the proportion of the intersecting area to the smaller area of both areas to be matched. If the score is greater 1, the score -1 is assumed. This score will then be stored in the 'matching map' with both areas. This is iterated over all time points till `maxTime`.

Then, for `i = 0` the cell areas are initiated with `DynamicArea` and for time point zero a `CellStartEvent` is generated. For all `i>0` the predecessors of the area and the matching scores are called from the 'matching map' if the value is above the `intersectTolerance`. This values are subsequently grouped as follows:

- Cells with same predecessor: `CellSplitEvent`

- Cells without predecessor: `CellStartEvent`

- Cells with precisely one predecessor: `CellAliveEvent`

- Cells with more then one predecessor: `CellFusionEvent`

For each concatenation of cell areas, a 'lifeline' (`DynamicArea`) is created and all areas and respective time points added. `CellFusionEvents` and `CellSplitEvents` interrupt 'lifelines' and start new ones. After the whole stack is processed, post-processing starts. In this, `CellFusionEvents` are verified. If for the same cell area a `CellSplitEvent` occurs after a `CellFusionEvent`, it is with high fidelity a `CellTouchEvent`/`CellDeTouchEvent`. Thus, the 'lifeline' is corrected and the cell area with best matching score is used as predecessor. After `CellFusionEvent` correction, for all cells without successor a `CellEndEvent` is added. Furthermore variables 'birth' and 'death' events are set and the 'lifespan' calculated.

## Area calculation and filter design

The FS2.0 does not have a method implemented to automatically import filters. However, for area calculation and pre-processing filters, we used interface files and built the GUI using annotations and reflections. Thus, it is relatively easy to copy-paste an existing filter class, rename it, and adjust the conditions to create additional custom filters. This files just have to be added to the interface file and the `gui.model`.

An example using the `FilterLaPlace` class can be found in the S4 Fig in S1 File.

## Availability

The FS2.0 is available under the GNU Public License and can be used, modified and redistributed freely without warranty given by the developers. A version of the software, sources, tutorial, installation notes, and example data can be either obtained by the data package (DOI) associated with this paper or via our website www.filament-sensor.de. Please note that the software also contains the Focal Adhesion and Fiber Cross-correlation Kit (FAFCK) [6], as the development of both are closely intertwined. FAFCK uses the FS2.0 as internal package for fiber analysis before combining this with focal adhesion tracking over time and reliably providing cross-correlation statistics. Thus, we did not split the repository to not facilitate deviations between both projects. Both projects are however using seperated GUI classes and seperated test routines and data.

The FS2.0 has been tested on Windows and Linux. Using Java 8 the `.jar` file runs on click on windows. Running the FS2.0 on Java 9 is not advised. For Java 10 and above, Java does not contain the JavaFX package, which has to be installed separately and the path added. Please adhere to the installation notes to do so. Also, we advise to use `OpenJDK` and `OpenJFX` which both have to be installed and the path added on Windows and Linux. Detailed instructions how to do this can also be found in the installation notes given with the online repository DOI (https://doi.org/10.5281/zenodo.7314056) or on our website.

## Discussion and conclusions

In this paper we demonstrate major improvements for unsupervised and robust high-throughput fiber detection of our previously published FilamentSensor, a Java based software packet for detection of straight fibers with a basic GUI [22]. The major advancements of the new FilamentSensor2.0 are detection of curved fibers and the possibility to track individual fibers over time.

Furthermore, we added several plugins: brightness and contrast adjustment, bounding box for cell event detection, and a verification tool (against user drawn binary maps as well as

between different projects). We implemented all plugins and the improved GUI based on stack handling and viewing. This leads to improved usability and easier visual inspection of processing and detection results for all plugins. In parallel to stack handling implementation, we expanded the range of usable image formats. Saving and importing of settings is now possible as well as automated single filament tracking and area annotation in the batch analysis mode. For users with basic programming experience, new filter classes can easily be added in the code with our interface solution. Experienced programmers can use our fully documented code under a GNU Public License.

We created new ground truth data sets of cells stained for actin, myosin and vimentin to show the usability of our code for different fiber types in single and batch processing. Our software shows promising results for analysis of bulk statistics for whole cell populations as well as single fiber dynamics at the single cell level. In summary, the possibility to analyze all filamentous components of the cytoskeleton in an unbiased, robust and quantitative way offers new possibilities to study and analyze cytoskeletal structure formation and dynamics as well as potential crosstalk between the different elements to elucidate the complex interaction of cellular mechanics.

## Supporting information

**S1 File.**
(PDF)

## Acknowledgments

We thank Daniel Härtter for assistance with coding in Python.

## Author Contributions

**Conceptualization:** Florian Rehfeldt.

**Data curation:** Lara Hauke, Andreas Primeßnig, Jennifer Radwitz.

**Formal analysis:** Lara Hauke, Florian Rehfeldt.

**Funding acquisition:** Stefan F. Huckemann, Florian Rehfeldt.

**Investigation:** Lara Hauke, Jennifer Radwitz.

**Methodology:** Andreas Primeßnig.

**Project administration:** Florian Rehfeldt.

**Resources:** Stefan F. Huckemann, Florian Rehfeldt.

**Software:** Lara Hauke, Andreas Primeßnig, Benjamin Eltzner.

**Supervision:** Stefan F. Huckemann, Florian Rehfeldt.

**Validation:** Lara Hauke.

**Visualization:** Lara Hauke, Benjamin Eltzner, Jennifer Radwitz.

**Writing – original draft:** Lara Hauke, Benjamin Eltzner, Florian Rehfeldt.

**Writing – review & editing:** Lara Hauke, Andreas Primeßnig, Benjamin Eltzner, Jennifer Radwitz, Stefan F. Huckemann, Florian Rehfeldt.

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
