## [Decision Letter · Decision Letter 0]

29 Aug 2022

PONE-D-22-21354FilamentSensor 2.0: An open-source modular toolbox for 2D/3D cytoskeletal filament trackingPLOS ONE

Dear Dr. Rehfeldt,

Thank you for submitting your manuscript to PLOS ONE. After careful consideration, we feel that it has merit but does not fully meet PLOS ONE’s publication criteria as it currently stands. Therefore, we invite you to submit a revised version of the manuscript that addresses the points raised during the review process.

Both Reviewers provided very constructive feedback on your manuscript that should substantially improve readability and impact of this work. Please review the Reviewers’ comments carefully and address them in revised manuscript. The Reviewers asked to include additional references and details on the methods, remove ambiguous statements, add the summary table describing the functionality of this software, describe the main advantages and disadvantages of the software in respect to other available tools.

We look forward to receiving your revised manuscript.

Kind regards,

Yulia Komarova

Academic Editor

PLOS ONE

Journal Requirements:

6. We note you have included a table to which you do not refer in the text of your manuscript. Please ensure that you refer to Table 1 in your text; if accepted, production will need this reference to link the reader to the Table.

Reviewers' comments:

Reviewer's Responses to Questions

**Comments to the Author**

1. Is the manuscript technically sound, and do the data support the conclusions?

Reviewer #1: Yes

Reviewer #2: Partly

2. Has the statistical analysis been performed appropriately and rigorously? 

Reviewer #1: Yes

Reviewer #2: No

3. Have the authors made all data underlying the findings in their manuscript fully available?

Reviewer #1: Yes

Reviewer #2: Yes

4. Is the manuscript presented in an intelligible fashion and written in standard English?

Reviewer #1: Yes

Reviewer #2: Yes

5. Review Comments to the Author

Reviewer #1: In this manuscript, Hauke et al. present an upgrade to their previous FilamentSensor pipeline: FilamentSensor 2. This software addresses how to identify and track filaments in typical imaging experiments of fluorescently labeled cytoskeletal filaments, e.g. actin or microtubules. This is an important experimental problem for which good software solutions are critical. Importantly, this work provides a complete ImageJ plugin complete with a GUI which will greatly enhance usability. In this upgraded version of FilamentSensor, real-time tracking can be performed including on filaments with curvature, which was missing in the previous version.

I think the method presented in this work is new, important, and well documented, and I would therefore recommend publication.

I do not have any major comments that would need to be addressed, only a few minor comments:

- I think the introduction could be improved for readability and flow and in particular for better highlighting the edge of this method over other methods. In lines 30-33 it does not really become clear what is missing in previous methods that needs to be addressed. A clear list of the specific things that are new in this software version early on could help.

- the summary in the introduction "It can extract all relevant data for all cellular filaments" is clearly an overstatement that is not informative and should be revised.

- sometimes not enough references are given. E.g. line 45

"While many scripts and packages exist and are publicly available for different approaches on fiber tracking, these are often not accessible for scientists without programming experience. " - some specific references here would be appropriate

- the references are confusingly numbered

Reviewer #2: In this manuscript, the authors describe the updated version 2.0 of their FilamentSensor, a software tool for automated cytoskeletal filament tracking and analysis. The structure and dynamics of the cytoskeleton is central for many cellular functions, and quantitative analysis of CSK filaments from microscopy images can be a powerful technique with many applications in biology. The previously published versions of FilamentSensor already have found a number of useful applications, as evident in the number of citations. It is great that it continues to be developed and updated, and the amount of improvements in this version together with the availability of new benchmark data justifies another journal publication in my opinion. There are some shortcomings in the way the work is presented that should be addressed before publication, as summarized below:

1) could you add more details about how the benchmark data were acquired, apart from the gel stiffness? Especially imaging parameters, microscopy modalities, stains used etc. would be helpful

2) a discussion of the robustness of the method with regards to data quality and imaging modalities would be great, so that readers can judge if the tool can be applied for their specific purpose

3) the presentation of the quantitative results is less than optimal - the boxplots do not give a good impression of the distributions (scatterplots would be better), and a simple statistical analysis should be included

4) the improvements compared to previous versions are described in the text but it is not easy to get a good overview this way - a table summarizing the additional functionality and parameters, as the authors already prepared for the performance improvements (Table 1), would be great

5) a comparison to existing tools is missing, at least in the discussion the authors should briefly mention the main advantages and disadvantages of their software compared to other published work

6) including a code example is very helpful, but the way it is inserted into the main manuscript disrupts the text in my opinion - maybe include as separate box or move to supplement?

7) what is "FAFCK" mentioned in the Availability section?

Finally, to increase the possibilities for interaction with users of the FilamentSensor, did the authors consider to become a community partner of the image.sc forum?

6. PLOS authors have the option to publish the peer review history of their article (what does this mean?). If published, this will include your full peer review and any attached files.

Reviewer #1: No

Reviewer #2: No

---

## [Author Response · Author response to Decision Letter 0]

15 Nov 2022

Response Letter to the Reviewers

Reviewer’s Comments in black 

Our response in blue

Reviewer #1: In this manuscript, Hauke et al. present an upgrade to their previous FilamentSensor pipeline: FilamentSensor 2. This software addresses how to identify and track filaments in typical imaging experiments of fluorescently labeled cytoskeletal filaments, e.g. actin or microtubules. This is an important experimental problem for which good software solutions are critical. Importantly, this work provides a complete ImageJ plugin complete with a GUI which will greatly enhance usability. In this upgraded version of FilamentSensor, real-time tracking can be performed including on filaments with curvature, which was missing in the previous version.

I think the method presented in this work is new, important, and well documented, and I would therefore recommend publication.

We are greatly appreciate the reviewer’s enthusiastic and positive comments and the recommendation for publication. We addressed the few minor comments in the following. 

I do not have any major comments that would need to be addressed, only a few minor comments:

- I think the introduction could be improved for readability and flow and in particular for better highlighting the edge of this method over other methods. In lines 30-33 it does not really become clear what is missing in previous methods that needs to be addressed. A clear list of the specific things that are new in this software version early on could help.

We edited the introduction to improve the readability and to clarify the improvements of the FS2.0 in comparison to the previous version. We added Table 1 for a quick overview (see also response to reviewer #2). However, a more detailed comparison to state of the art software for fiber detection is out of scope of this paper as there is a huge variety of programming environments, 2D/3D data input and desired features, e.g. node or density analysis.

- the summary in the introduction "It can extract all relevant data for all cellular filaments" is clearly an overstatement that is not informative and should be revised.

We agree with the reviewer and revised the phrase accordingly.

- sometimes not enough references are given. E.g. line 45

"While many scripts and packages exist and are publicly available for different approaches on fiber tracking, these are often not accessible for scientists without programming experience. " - some specific references here would be appropriate

We added some specific references in the appropriate section (Refs. 15-19). 

- the references are confusingly numbered

Thank you very much for noting, we changed to the appropriate reference style. 

Reviewer #2: In this manuscript, the authors describe the updated version 2.0 of their FilamentSensor, a software tool for automated cytoskeletal filament tracking and analysis. The structure and dynamics of the cytoskeleton is central for many cellular functions, and quantitative analysis of CSK filaments from microscopy images can be a powerful technique with many applications in biology. The previously published versions of FilamentSensor already have found a number of useful applications, as evident in the number of citations. It is great that it continues to be developed and updated, and the amount of improvements in this version together with the availability of new benchmark data justifies another journal publication in my opinion.

We are very happy and thankful for the reviewer’s positive feedback and the recommendation for publication after addressing the comments on presentation. 

There are some shortcomings in the way the work is presented that should be addressed before publication, as summarized below:

1) could you add more details about how the benchmark data were acquired, apart from the gel stiffness? Especially imaging parameters, microscopy modalities, stains used etc. would be helpful

We added detailed information on methods for the benchmark dataset.

2) a discussion of the robustness of the method with regards to data quality and imaging modalities would be great, so that readers can judge if the tool can be applied for their specific purpose

We fully agree that more information and discussion will be helpful for the readers. We therefore added the new subsection ‘Applicability’ according to the reviewer’s suggestion.

3) the presentation of the quantitative results is less than optimal - the boxplots do not give a good impression of the distributions (scatterplots would be better), and a simple statistical analysis should be included

We are thankful for this suggestion and changed the boxplots in Figures 2, 3, 4 and S1 to scatterplots for a better presentation of the data. Where appropriate, we performed Kolmogorov-Smirnov-tests and added the information in the captions.

4) the improvements compared to previous versions are described in the text but it is not easy to get a good overview this way - a table summarizing the additional functionality and parameters, as the authors already prepared for the performance improvements (Table 1), would be great

We greatly appreciate this suggestion and added Table 1 for a quick overview of the improvements in the FS 2.0 (see also Reviewer #1).

5) a comparison to existing tools is missing, at least in the discussion the authors should briefly mention the main advantages and disadvantages of their software compared to other published work

We fully acknowledge the reviewer’s request for a comparison to existing tools (see also responses 1 & 3 to Reviewer #1). We edited the introduction to clarify the improvements of the FS2.0 in comparison to the previous version (and added Table 1) and in contrast to other existing software. However, a more detailed comparison is out of scope of this paper as there is a huge variety of programming environments, 2D/3D data input and desired features, e.g. node or density analysis. We also added some specific references in the appropriate section (Refs. 15-19).

6) including a code example is very helpful, but the way it is inserted into the main manuscript disrupts the text in my opinion - maybe include as separate box or move to supplement?

We do agree with the referee thank for the suggestion. Accordingly, we moved the code sample to supplemental information.

7) what is "FAFCK" mentioned in the Availability section?

We added more details to the FAFCK (Focal Adhesion Filament Cross-correlation Kit) in the Availability section. 

Finally, to increase the possibilities for interaction with users of the FilamentSensor, did the authors consider to become a community partner of the image.sc forum?

Thank you very much for the suggestion to become a community partner of the image.sc forum. We will definitely consider this for a wider dissemination of our toolbox.

---

## [Decision Letter · Decision Letter 1]

6 Dec 2022

FilamentSensor 2.0: An open-source modular toolbox for 2D/3D cytoskeletal filament tracking

PONE-D-22-21354R1

Dear Dr. Rehfeldt,

We’re pleased to inform you that your manuscript has been judged scientifically suitable for publication and will be formally accepted for publication once it meets all outstanding technical requirements.

Kind regards,

Yulia Komarova

Academic Editor

PLOS ONE

Additional Editor Comments (optional):

Reviewers' comments:

Reviewer's Responses to Questions

**Comments to the Author**

1. If the authors have adequately addressed your comments raised in a previous round of review and you feel that this manuscript is now acceptable for publication, you may indicate that here to bypass the “Comments to the Author” section, enter your conflict of interest statement in the “Confidential to Editor” section, and submit your "Accept" recommendation.

Reviewer #1: All comments have been addressed

Reviewer #2: All comments have been addressed

2. Is the manuscript technically sound, and do the data support the conclusions?

Reviewer #1: Yes

Reviewer #2: Yes

3. Has the statistical analysis been performed appropriately and rigorously? 

Reviewer #1: Yes

Reviewer #2: Yes

4. Have the authors made all data underlying the findings in their manuscript fully available?

Reviewer #1: Yes

Reviewer #2: Yes

5. Is the manuscript presented in an intelligible fashion and written in standard English?

Reviewer #1: Yes

Reviewer #2: Yes

6. Review Comments to the Author

Reviewer #1: The authors have addressed my comments, in particular on improving the introduction and referencing in the article. I'm therefore happy to support publication.

Reviewer #2: The authors have greatly improved their manuscript, eliminating all weak points in my opinion. I have no further questions.

7. PLOS authors have the option to publish the peer review history of their article (what does this mean?). If published, this will include your full peer review and any attached files.

Reviewer #1: No

Reviewer #2: No
